# VIIRS Nighttime Lights in the Estimation of Cross-Sectional and Time-Series GDP

**Xi Chen** [1,*] **and William D. Nordhaus** [2]

[1]    Department of Sociology, Quinnipiac University, Hamden, CT 06518, USA

[2]    Department of Economics, Yale University, New Haven, CT 06511, USA; william.nordhaus@yale.edu

*    Correspondence: xi.chen@quinnipiac.edu; Tel.: +1-203-582-6408

**Abstract:** This study extends previous applications of DMSP OLS nighttime lights data to examine the usefulness of newer VIIRS lights in the estimation of economic activity. Focusing on both US states and metropolitan statistical areas (MSAs), we found that the VIIRS lights are more useful in predicting cross-sectional GDP than predicting time-series GDP data. This result is similar to previous findings for DMSP OLS nighttime lights. Additionally, the present analysis shows that high-resolution VIIRS lights provide a better prediction for MSA GDP than for state GDP, which suggests that lights may be more closely related to urban sectors than rural sectors. The results also indicate the importance of considering biases that may arise from different aggregations (the modifiable areal unit problems, MAUP) in applications of nighttime lights in understanding socioeconomic phenomenon.

**Keywords:** VIIRS light; cross-sectional; time series; GDP; economic statistics

## 1. Introduction

The digital data archive of nighttime lights was first established at the National Oceanic and Atmospheric Administration (NOAA) National Geophysical Data Center (NGDC) in 1992. To date, the archive contains lights products based on two sensors—the early instrument of the U.S. Air Force Defense Meteorological Satellite Program-Operational Linescan System (DMSP-OLS), and the current NASA/NOAA SNPP satellite's the Visible Infrared Imaging Radiometer Suite (VIIRS) [1].

Studies focusing on the application of nighttime lights have flourished since the publication of annual stable lights based upon DMSP-OLS by the NOAA Earth Observation Group (EOG) [2]. Following this initial publication of data, NOAA EOG has published several other light data products, including intercalibrated lights for multiple years [2]. The public accessibility and user-friendly features of these data have benefited scholars around the world, aiding in the advancement of human-activity-related studies in many disciplines.

In the last few years, the EOG has processed and published nighttime lights from the VIIRS [3], which flown on the satellite launched in 2011 [1]. Some new features of VIIRS lights data include monthly updates, higher resolution, and better overall quality. With these new features, the VIIRS lights has the potential to create another increase in nighttime lights-based studies focusing on social and economic activities.

Testing the applicability of new VIIRS lights in estimating of economic statistics provides an important initial step. It is well known that economic development significantly influences and is highly correlated with inequality, poverty, public health, migration, urbanization, and a wide range of social phenomenon. Therefore, understanding the relationship between nighttime lights and economic statistics, such as gross domestic product (GDP), could shed new light on the connections between nighttime lights and patterns or changes in other social institutions or individual behaviors.

The inquiry of the GDP-light association started when DMSP-OLS lights first became available. In the last decade, over a thousand publications have used DMSP-OLS lights to predict economic activities. Elvidge et al [4], Elvidge et al [5], Doll et al. [6], Chen and Nordhaus [7], Henderson, Storeygard, Weil [8,9], and Ghosh et al. [10] pioneered research estimating socio-economic statistics using nighttime lights. These studies were important because, unlike conventional economic statistics provided by governments or international organizations, which are often inconsistent in timeframe, definition, and measurement instrument employed, nighttime lights have the advantage of being objectively measured, updated regularly, and cover the entire globe (except for at high latitudes). Additionally, lights data avoid systematic errors due to government misreporting or differences in methods, and this may make them a more reliable source in predicting true GDP values at the global, national, and local levels.

The availability of DMSP-OLS annual stable lights from 1992 to 2013 allows researchers to investigate the GDP-lights relationship with cross-sectional and time series data during 1990s and 2010s. In general, the results suggest the DMSP-OLS lights are useful for cross-sectional analysis, but it may not provide enough information to predict time series economic statistics [11]. Henderson et al. [8] found that the elasticity on overall growth of GDP with respect to stable light is close to 0.3, while separating agricultural and non-agricultural sectors (see [12]) suggested that the increase in luminosity at night is largely the result of expansion in non-agricultural economic activity, and lights do not explain value-added GDP in agriculture and forestry.

The availability of VIIRS data with its new features offers another opportunity to reexamine the light-GDP relationship, and therefore to continue the previous analysis and expand it with data collected after 2012. The current VIIRS data have been updated to 2018, and is available monthly and annually for multiple years. Our goal is to test the GDP-lights association using the new VIIRS lights data. The focus is US GDP statistics, as these are relatively accurate account of economic output compared to the statistics reported in many other countries. We use annual level GDP in cross-sectional analysis and 1- and 2-year GDP growth rates in time-series analysis.

## 2. Materials and Methods

### 2.1. Literature and Background

Past research consistently shows that DMSP nighttime lights products, particularly stable lights, provide useful information for economic statistical estimates, at least for cross-sectional level measures [7–9,13,14]. Specifically, Nordhaus and Chen [7] proposed an optimal weighting approach to improve GDP estimates at the regional and grid cell levels and found that stable lights added more value for GDP estimates of the poorest countries, those that received D or E grade ranking (similar to the grading system of the Penn World Table 6.3), but added very little information for wealthy or middle-income countries (A, B, and C grades). In addition, they also found that the information provided by DMSP stable lights are more useful in estimating cross-sectional GDP than GDP growth, and such results were observed at the national level as well as at the 1° latitude by 1° longitude grid cell output generated by the Yale GEcon project (gecon.yale.edu).

Following these initial findings, Nordhaus and Chen used bootstrap analysis to test the uncertainty in estimating GDP with the stable light [14]. They again found observable difference between cross-sectional and time-series data. The uncertainty in light-based GDP estimates are much larger in time series than in cross-sectional data. For cross-sectional estimates, the uncertainty in optimal estimates of GDP depends primarily on the measurement errors in the conventional GDP estimates, but less on the measurement errors in nighttime lights. This important finding implies the lights-GDP association is most likely country/regional-specific, because the quality of conventionally-measured GDP varies substantially by countries.

With the available VIIRS lights, more studies have started to explore lights-related research topics with VIIRS. Yet, only a small number of them conducted comparative studies of DMSP-OLS stable

and VIIRS lights, which is likely due to the fact that the two lights products overlap only by 2 years (2012–2013). One of these comparative studies used the early products of the VIIRS and stable lights in estimating population and economic statistics for Sub-Sahara Africa [11]. Because the early products of VIIRS lights only contained the two months composite image of VIIRS day/night band (DNB) of observations during 4–18 April 2012 and 11–23 October 2012, the study on Sub-Sahara Africa tested the cross-sectional data for 2012. The results showed that high quality VIIRS lights have higher correlation with grid population and grid cell economic output compared to results based on stable lights [11]. Their analysis also indicated that VIIRS lights data are especially useful for estimating population and output in regions with extremely low economic density, as DMSP stable lights are often coded as no value in such areas, but VIIRS can detect low lights and assign values to those locations [11]. Thus, using VIIRS lights can add more observations in the sample than using DMSP stable lights, since almost one-third of grid-cells with positive population and output in the GEcon data set were coded as zero in the stable lights. Often in regression analysis, zero-value observations are automatically deleted from the sample in logarithmic transformation as a standard procedure, which leads to a large reduction of sample size, especially in studies for less developed or remote regions.

VIIRS lights are still not ideal as predictors for population or economic statistics. It has been shown that VIIRS light explain only 30% to 40% of the variability of output and population for Sub-Sahara Africa [11]. Studies that have focused on predicting the rapid economic growth in China have also compared stable light and VIIRS, including Jing et al. [15], Dai and Zhao [16], Li et al. [17] and Shi et al. [18]. Using Chinese regional economic statistics, these studies all concluded that VIIRS has better predictive power for estimating output than other DMSP-OLS data. More recent studies have examined the cross-sectional relationship between VIIRS lights and GDP statistics, and they found that, similar to the results of DMSP stable lights, there is strong correlation between VIIRS and GDP [19,20].

The better results of the VIIRS-GDP association can be explained by the improved quality with the VIIRS instrument, the Suomi National Polar Partnership (SNPP) satellite launched in 2011. The pixel footprint of VIIRS data are 742 m × 742 m. By contrast, the pixel footprint of DMSP OLS data are 5 km × 5 km (these measurements are at the nadir and expand at the edge of scan). Thus, the pixel ground footprint of the VIIRS image is 45 times smaller than the DMSP footprint, which leads to high spatial resolution VIIRS lights data; it can therefore be stored in 15 arc-seconds [21,22]. In addition, with wider radiometric detection range and onboard calibration, VIIRS light images has no saturation problems [16], which is a serious concern for researchers when they estimate economic statistics with DMSP stable lights. Saturation occurs because the digital values of lights are top-coded at 63, which cannot distinguish in populated regions the most productive area from the relatively less productive ones.

Additionally, with 22 spectral bands and equipped with onboard calibration, the VIIRS instrument generates lights images based upon more information than DMSP lights. For instance, its Day/Night band (DNB) has a lower detection threshold which can record dimmer lights [21,22]. This allows VIIRS lights to capture activities at regions with low-population density that cannot be detected in DMSP OLS. The superior features of VIIRS lights suggest it may be more applicable in estimating economic statistics, compared to early DMSP OLS lights products.

Over the last few years, the EOG produced multiple lights products based on VIIRS. The early composite images were based on observations of limited days, such as April 4th to 18th in 2012. Recently, the EOG started publishing VIIRS lights on a monthly basis, and also updated annual composites for multiple years. These new data allow us to test the relationship between lights and economic statistics in both cross-section and time series. We undertake this task as a continuation of previous efforts to test the stable light-GDP association in the hope of shedding improved insight into the relationship between lights and economic activities.

*2.2. Data and Methods*

To test the association between VIIRS light and GDP, we use VIIRS light to predict 2014 to 2016 real GDP (as chained 2009 dollars) at both the state and metropolitan statistical area (MSA) level. The U.S. Office of Management and Budget (OMB) defines a metropolitan statistical area (MSA)as a core-based statistical area (CBSA) associated with at least one urbanized area with a population of at least 50,000. The GDP data by state and MSA are downloaded from the official website of the U.S. Bureau of Economic Analysis. The state and MSA samples include annual GDP for 50 states plus the District of Columbia and of 381 MSAs. For VIIRS lights, we use annual data for the corresponding time periods, i.e. annual average lights for 2014–2016. The initial publication of VIIRS only included image composites for April and October 2012. The updated VIIRS lights include monthly composites from April 2014 to the present, and the annual composites for 2015 and 2016, but not for 2014. To generate consistent measure of annual VIIRS lights, we use averaged monthly composites as an annual light measure for 2014, 2015, and 2016. The annual composites of 2015 and 2016 are also tested as comparisons to the results based on monthly composites. Both monthly and annual composite images (Version 1) are produced by the Earth Observations Group (EOG) at the National Oceanic and Atmospheric Administration (NOAA). The steps generating this version of VIIRS images can be found in Elvidge et al. [23]. The temporal lights, such as those from aurora, fires, boats, and other background noises are filtered out in the annual composites, but not in the monthly composites. In other words, the annual composites of VIIRS lights are better representation of stable lights.

The pixel value of the VIIRS is a non-integer, ranging from around −1.0 to around 8000.0 within U.S. terrestrial boundaries in the February monthly file. Note that the VIIRS sensor has a noise floor, in which the digital values can be a very small negative value (See Chen and Nordhaus 2015 for a detailed discussion) [11]. The units of VIIRS images are nanowatts/cm2/sr, and its digital values are stored in 15 arc-second grids. Even with high resolution, lights are influenced significantly by stray lights. In general, stray lights affect 25% of nighttime scenes [24]. Thus, we use the stray-light-corrected version of data, which is only available after 2014. The digital values of VIIRS lights are aggregated at the state and metropolitan levels and then merged with real GDP by the administrative unit.

We use linear regression to estimate the relationship between VIIRS and state and MSA GDP (both are measured logarithmically to correct for heteroskedasticity). Our goal is to test whether the VIIRS lights are more closely associated with cross-sectional or time series GDP statistics. This question is especially important, as previous studies have suggested that the early nighttime lights (stable lights) strongly correlate with cross-sectional data but poorly correlate with time series data. Yet, these studies only examine DMSP OLS lights, which are affected by saturation, missing data in low density areas, and calibration problems. Current literature that examines the application of VIIRS lights in estimation of economic statistics only uses cross-sectional VIIRS data. As such, this is the first study to use the VIIRS to predict both cross-section and time series economic statistics.

## 3. Results

First, we compare average monthly composites and annual composites lights at the state (Figure 1a) and the MSA levels (Figure 1b). In general, average monthly composite and annual composite lights have a similar distribution. Alaska is an obvious outlier, probably because its monthly composite lights are heavily influenced by the temporal lights of aurora and fishing boats. The two metropolitan areas in Alaska, Anchorage and Fairbanks, are also outliers in the MSA scatter plot, yet to a much lesser degree.

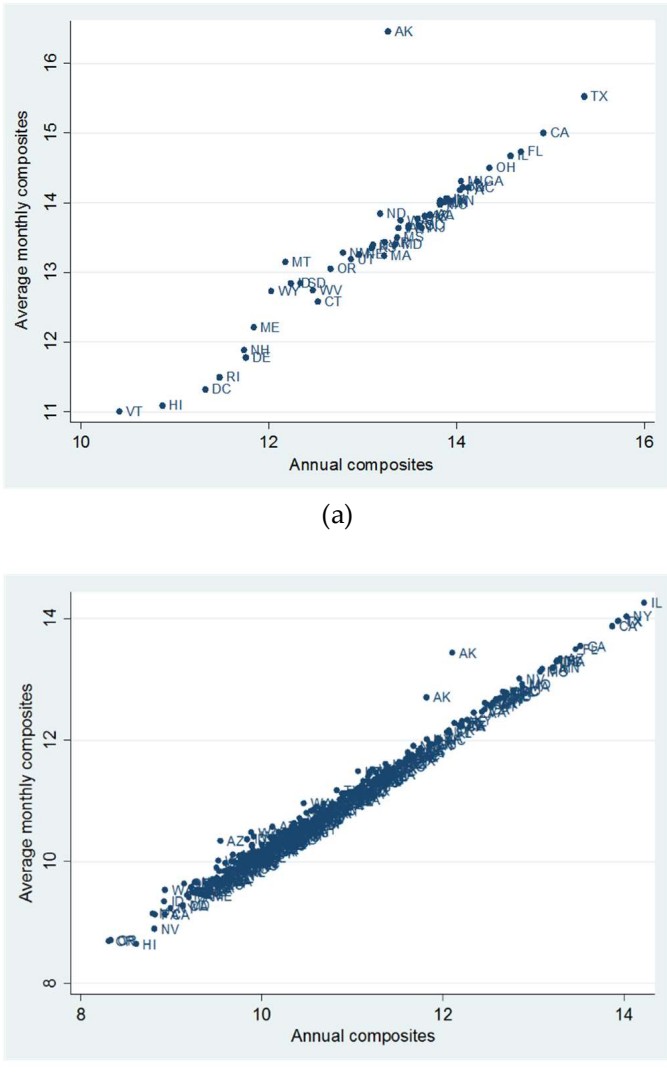

**Figure 1.** Average monthly composite and annual composite lights for states (2016) (**a**) and MSAs (2016) (**b**).

Our main analysis uses average monthly composite lights of 2014, 2015 and 2016, and annual composite lights of 2015 and 2016. We start with state level analysis. Figure 2 illustrates differences in the prediction of annual GDP and GDP growth rates with scatterplots. For 2016, there is a relatively strong correlation between annual lights and GDP, except for Alaska (Figure 2a). Yet, there is virtually no relationship between the two-year growth rate in lights and GDP in the plot (Figure 2b). In a preliminary ordinary least squares regression analysis (Figure 2), the absolute residual for Alaska is around 4.0 in predicting annual GDP. Because of Alaska's large effect on the fitted regression line using average monthly composite lights, Alaska is excluded from the sample in this part of analysis. We keep Alaska in the sample in the annual composite lights analysis. The robust regressions are used in all following analysis to address the model sensitivity to outlier issue.

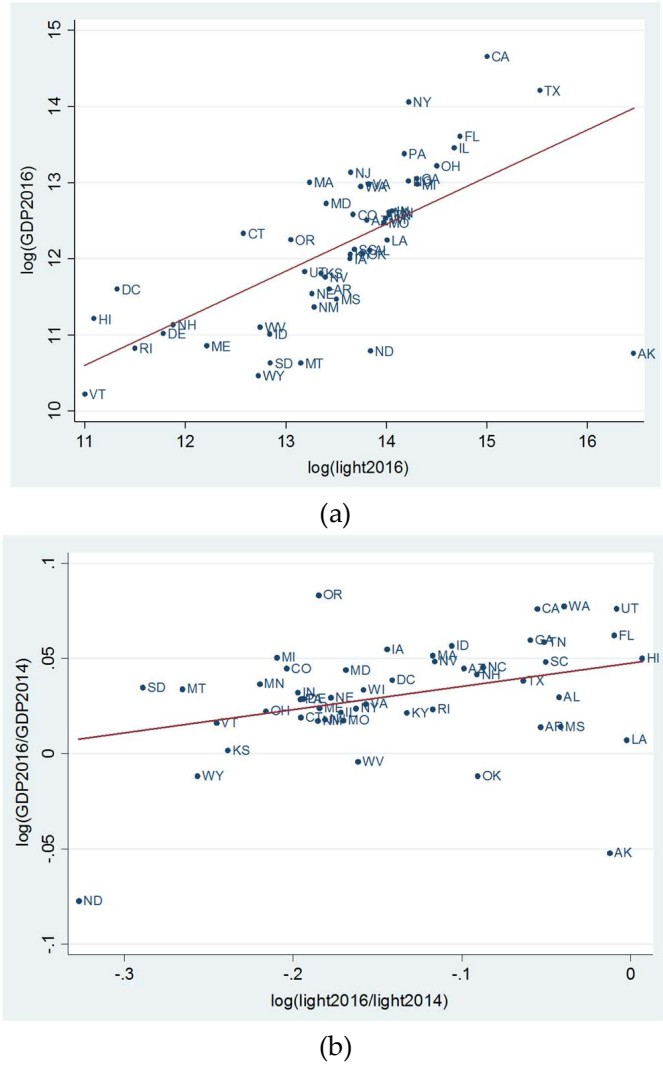

**Figure 2.** State lights (average monthly composites) and GDP annual (2016) (**a**) and 2-year growth (2014–2016) (**b**) data.

Table 1 column 1–3 report robust regression coefficients and standard errors for pooled annual data, and for 1-year and 2-year growth rates, using average monthly composites. Year effects are controlled in all models. The simple comparison of the coefficients indicates that VIIRS lights has much stronger predictability for cross-sectional GDP than for GDP growth. For the U.S. states, lights can explain about 61% of variance in cross sectional GDP, but only 13–18% of variance for one-year or two-year growth rates. The coefficient of lights in predicting annual GDP (0.84) is also much larger than those in predicting GDP growth rates.

We run the similar analysis with annual composite lights and report these results in Table 1 columns 4–5. There are no substantive changes in the coefficients of lights and light growth rates. Again, the coefficient of lights in predicting annual GDP is much higher (around 0.87) than the coefficient in predicting GDP growth rates (0.03). Alaska is included in testing annual composite lights, as it does not appear to be an outlier nor change the estimated results significantly.

**Table 1.** Robust regression results for state annual and growth data.

| | Log(GDP) | Log(GDP$_{t+1}$/GDP$_t$) | Log(GDP$_{2016}$/GDP$_{2014}$) | Log(GDP) | Log(GDP$_{2016}$/GDP$_{2015}$) |
|---|---|---|---|---|---|
| Independent variable | | Average monthly composite lights | | | Annual composite lights |
| Log(light) | 0.843 *** | | | 0.872 *** | |
| | (0.055) | | | (0.055) | |
| Log(light$_{t+1}$/light$_t$) | | 0.038 * | | | 0.031 |
| | | (0.018) | | | (0.021) |
| Log(light$_{2016}$/light$_{2014}$) | | | 0.125 ** | | |
| | | | (0.037) | | |
| Year 2015 | 0.087 | | | | |
| | (0.129) | | | | |
| Year 2016 | 0.144 | −0.009 *** | | 0.070 | |
| | (0.129) | (0.003) | | (0.112) | |
| Constant | 0.719 | 0.025 *** | 0.052 *** | 0.555 | 0.016 *** |
| | (0.751) | (0.002) | (0.006) | (0.739) | (0.002) |
| N | 150 | 100 | 50 | 102 | 51 |
| adj. R-sq | 0.610 | 0.128 | 0.180 | 0.709 | 0.023 |
| SER | 0.4173 | 0.0002 | 0.0004 | 0.3181 | 0.0002 |

Robust regression coefficients and standard errors in parenthesis. * $p < 0.05$, ** $p < 0.01$, *** $p < 0.001$. The reference year is year 2014 in column 1, and it is year 2015 in column2 and column4.

Next, we examine the metropolitan data. Table 2 column 1–4 shows the results for average monthly composite lights. Similar to results of the state analysis, the cross-sectional, annual data have much better results than the growth data. Specifically, lights predict about 89% of the variance in annual MSA GDP, but only 2% of the variance in MSA GDP growth. Because the MSA sample is much larger than the state sample, we were able to test a subsample of MSAs, which includes two-year growth rates of 98 MSAs from the five large states, CA, TX, NY, MI, and FL. As the population, area size, and industry makeups tend to vary substantially by states, we want to examine lights-GDP relationship within larger states, and then compare these results with the results of the national sample. The model fit is noticeably better for large states: the coefficient of lights variable improved from 0.077 to 0.222, and variance in GDP growth explained increased from 2% to 20%. However, the improved growth rate results with the MSA subsample are still poor compared to the cross-sectional data. Figure 3's scatter plots illustrate the lights-GDP association at the MSA level. The plot in the top panel depicts a very strong cross-sectional correlation for 2016 annual lights and GDP for MSAs, while the plot in the bottom panel shows no relationship between growth in lights and growth in GDP in U.S. MSAs. The results based on annual composite lights (Table 2 column 5~6) show the very similar results as those of average monthly composite lights.

**Table 2.** Robust regression results for metropolitan level and growth data.

| | Log(GDP) | Log(GDP$_{t+1}$/GDP$_t$) | Log(GDP$_{2016}$/GDP$_{2014}$) | | Log(GDP) | Log(GDP$_{2016}$/GDP$_{2015}$) |
|---|---|---|---|---|---|---|
| | | | All MSAs | MSAs in CA, TX, NY, MI, FL | | |
| Independent variable | | Average monthly composites lights | | | Annual composites lights | |
| Log(light) | 1.132 *** | | | | 1.073 *** | |
| | 0.012 | | | | 0.014 | |
| Log(light$_{t+1}$/light$_t$) | | 0.031 ** | | | | 0.045 *** |
| | | 0.011 | | | | 0.012 |
| Log(light$_{2016}$/light$_{2014}$) | | | 0.077 ** | 0.222 *** | | |
| | | | 0.025 | 0.045 | | |
| 2015 | 0.0801 ** | | | | | |
| | 0.030 | | | | | |
| 2016 | 0.131 *** | −0.007 *** | | | 0.080 * | |
| | 0.030 | 0.002 | | | 0.031 | |
| CONSTANT | −2.983 *** | 0.020 *** | 0.038 *** | −2.116 *** | 0.016 *** | |
| | 0.132 | 0.001 | 0.003 | 0.005 | 0.157 | 0.001 |
| N | 1143 | 762 | 381 | 98 | 762 | 381 |
| adj. R-sq | 0.887 | 0.022 | 0.022 | 0.197 | 0.879 | 0.032 |
| SER | 0.1731 | 0.0006 | 0.0019 | 0.0013 | 0.1840 | 0.0005 |

Robust regression coefficients and standard errors in parenthesis. * $p < 0.05$, ** $p < 0.01$, *** $p < 0.001$. The reference year is year 2014 in column 1 and 5, and it is year 2015 in column 2 and 6.

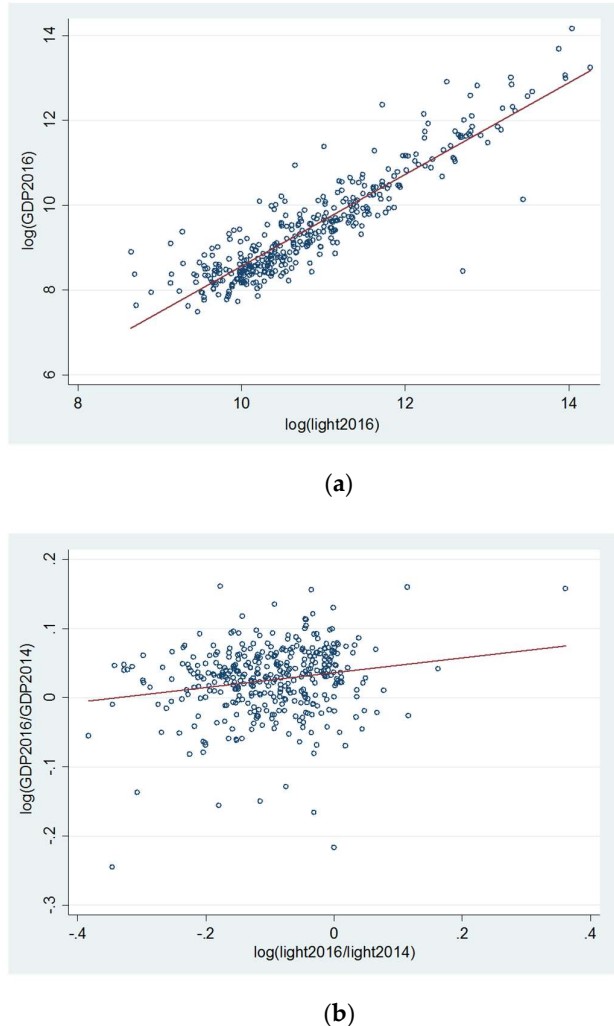

(**a**)

(**b**)

**Figure 3.** MSA lights (average monthly composites) and GDP annual data (2016) (**a**) and for 2-year growth (2014–2016) (**b**).

## 4. Discussion and Conclusion

Using the most recent VIIRS lights and US state and metropolitan GDP to test the lights-GDP relationship, we find that VIIRS lights data is useful to estimate cross-sectional GDP. This result is similar to what was found in DMSP stable lights. There are also new findings regarding the light-GDP relationship at different geographic units.

First, although VIIRS lights incorporate improved data quality and new features that are not present in DMSP OLS stable lights, specifically in terms of high-resolution, onboard calibration, Day/Night Bands to detect dimmer lights, and stray-light correction, it cannot yet be used to predict economic growth usefully, at least for regions within the United States. This is consistent in using different VIIRS products, including monthly composite and annual composite images.

Second, the association between lights and economic data is stronger at the metropolitan (MSA) level than at the state level. This pattern is particularly shown in cross-sectional data. At the state level, the coefficient of lights on annual GDP is around 0.85, while it is consistently close to 1.00 for MSAs. The standard error of regression (SER) also indicates that model fits the MSAs data (around 0.17) better than the state data (0.32–0.42). This is a prime example of the modifiable areal unit problem (MAUP). That is, the results are sensitive to the changes in scale of analysis [25]. Researchers who focus on applications of nighttime lights should be careful to consider the possible bias due to MAUP. The versatility of nighttime lights allow them to be aggregated within almost any geographic or

political boundaries as specified by a research topic, but this advantage many introduce biases. As the current study has shown, the results vary depending on which administrative unit is utilized.

There are two possible reasons why the nighttime lights-GDP correlation is stronger at the MSA level than at the state level. First, it is possible MSA GDP are estimated with less error, while the estimation of agricultural, fishing, forestry, and mining industries (heavily concentrated in rural areas) may include a larger error. Thus, at the MSA level, the correlations could appear stronger. Based on our knowledge of the disaggregated data, we suspect that the results do not arise from differences in the quality of output data.

A second and more likely reason is that the types of economic activity primarily occurring in MSAs, such as service, retail, transportation, etc., are more likely to be captured by nighttime lights, while the economic activity in non-MSA areas are mostly in industries less dependent on electric power for lighting at night, such as agriculture, mining, or forestry. Just as shown in Keola, Andersson, and Hall [12], changes in DMSP lights do not correspond to GDP growth in agriculture and forestry. Therefore, when GDP are calculated at the state level, including the rural industries, it may show a relatively weaker correlation with lights than MSA GDP.

The reasons for the weak association between lights and GDP in growth data remains a puzzle. One possibility is that errors or inconsistencies in digital images of luminosity captured by satellites over time. Even with onboard calibration, measurement error in VIIRS imagery over time could be large. Furthermore, the optical attributes of sensor can decay over time, which also influences the reliability of radiance measure at nights. Additionally, the radiance captured by VIIRS instruments may include seasonal changes, just as the effect of stray light on high-latitude regions increases in summer. How the light-GDP relationship changes by season is another topic that has not been carefully investigated.

However, based on internal evidence and earlier studies, the most likely reason for the weak time-series relationship is that the variation in GDP growth rates is so much smaller than the cross-sectional GDP variation. For example, the standard deviation in the GDP growth rate across states from 2015 to 2016 was 6.6%, while the standard deviation in the log level of GDP was 122%. The small variation in GDP growth is also reflected in SER statistics in regressions. In the state data, the average distance from data points to regression line is extremely small (.0002–.0004), and the model only explains 13–18% of the total variation in GDP growth. That is, there are very little variations in 1- or 2-year growth rates that can be explained by nighttime lights. If the errors in the lights-GDP relationship are relatively high, those would lead to low ability to predict growth rates compared to annual GDP.

In sum, even though VIIRS lights produce higher quality data than products from DMSP OLS stable lights, its applications in economic statistics is still subject to limitations, particularly in economic forecasting or proxy studies. The encouraging fact is that VIIRS lights has similar or even better results in estimating cross-sectional economic statistics, compared to DMSP lights. This has significant implications for estimating the level of GDP or GDP per capita for small areas or remote regions where GDP is unknown or calculated with large error. With more lights data becoming available and at higher frequency (e.g., daily VIIRS lights), we can test the light-GDP relationship in a longer time frame and with more advanced methods (e.g., mixed-frequency models or spatial-temporal models). Particularly important would be to use the high-resolution VIIRS data for low-income and low-density regions with poor data quality, such as tropical Africa. As other human activities are often inseparable from economic activity, understanding the relationship of lights and economic statistics should be placed at the forefront of research on the applications of nighttime lights in the social sciences.

**Author Contributions:** X.C. and W.D.N. have contributed equally to conceptualization, methodology, formal analysis, data curation, and writing (including original draft preparation, review and editing).

**Funding:** This research received no external funding.

**Conflicts of Interest:** There is no conflict of interest.

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
