# Peer review of "VIIRS Nighttime Lights in the Estimation of Cross-Sectional and Time-Series GDP"

_remotesensing, doi:10.3390/rs11091057_

Round 1

Reviewer 1 Report

The authors have submitted a significant improved manuscript. All my former remarks were addressed in the revised version, in particular my concerns about not using the monthly time-series (instead of annual data). Methods and results are much clearer presented and discussed now. I recommend publishing their work in the present form.

Author Response

Thank you for your comments. 

Xi Chen

Reviewer 2 Report

First of all, I appreciate the effort of the authors on their work to facilitate the dataset for GDP research. This submitted manuscript describes their latest attempt to incorporate the latest development in NTL as a GDP describer. The authors had demonstrated the benefit of using DMSP-OLS NTL to estimate GDP changes, however with less success temporally than cross-sectionally. As the VIIRS-DNB NTL become more available, this work applies the same method used on DMSP-OLS NTL and demonstrates how VIIRS-DNB NTL is indifferent from its predecessor.

In this sense, this work does not bring any new insight regarding the relationship between NTL and GDP, despite this time VIIRS-DNB NTL enables the authors to carry out analysis with monthly composites. However, I failed to notice any new findings from this new data availability in this work.

The overall layout of the manuscript is fine, but the rendering of figures and tables, as well as their descriptions fall short to satisfactory, which will be listed in detail. I sincerely wish the authors can extend the scope of this work with additional ideas based on the new possibility the VIIRS-DNB NTL provides, on top of reworking on the display of results. At for current version of this manuscript, I do not think it is ready to be accepted for publishing.

Here are my listed comments:

[1] Abstract: "The results also ... socioeconomic phenomenon." This work does not provide enough evidence to support this conclusion.

[2] Line 91-93: The difficulty of studies to compare DMSP and VIIRS NTL is not because they are only overlapped by 2 years. It's originated from the sensor design and lack of on-board calibration of DMSP-OLS, making direct comparison of readings from the two sensors impossible.

[3] Line 120: I do not think DMSP-OLS has a foot print of 5km x 5km. For VIIRS-DNB, the foot print size is maintained at ~742m along the scan.

[4] Line 128: The 22 bands provided by VIIRS does not all contribute to NTL making. This sentence might be misleading.

[5] Line 171-173: Although I know DMSP NTL might not correlate to GDP well enough with time-series data, I noticed that in the previous works done by the authors intercalibration was not carried out for stable lights they used. I believe EOG has intercalibration coefficients available for DMSP stable lights. I suggest the authors contact EOG for further advise. Furthermore, have the authors considered radiance calibrated DMSP nighttime lights?

[6] Line 180 to 182: outliner -> outlier? temporal light -> emphemeral light?

[7] Figure 1: The labels of the data points does not help in any way. It will be better if a 1:1 line is provided.

[8] Line 200: What is the unit of the absolute residual?

[9] Figure 2: Why is there not a figure for 2014 GDP and NTL comparison? The regression result (slope, R^2) should be printed in the figure.

[10] Line 216: "Year effects are controlled in all models." What does this mean?

[11] Table 1 & 2: The layout is very unfriendly for readers. I cannot refer to any numbers given the column indication in the caption. What does the result without any asterisk mean? What does the Log(GDP(t+1)/GDP(t)) column imply? The description for the two tables in the text is very insufficient.

[12] Figure 3: Where is 2014 GDP and light comparison result? The regression result should be printed in the figures.

[13] Line 301-310: GDP growth is not proportional to light growth in several ways, and it also relates to the state of the region. For a developing region, small GDP growth can bring large installation in exterior light installation. On the contrary, for a developed region, even large GDP growth might show little increase in lighting, for it is already saturated. Furthermore, when GDP falls in a region, the light might not fall simultaneously for most of the exterior lighting are public infrastructures and could last longer than private installation. On the other hand, when GDP grows, especially as it is a result of public investment, the exterior lighting is likely to grow in the same pace as a result of infrastructure investing.

[14] Suggest also citing the word from EOG:

National trends in satellite-observed lighting 1992-2012, Global Urban Monitoring and Assessment through Earth Obaservation

Lighting Tracks Transition in Eastern Europe, Land-Cover and Land-Use Changes in Eastern Europe after the Collapse of the Soviet Union

Author Response

Please see the attached file for responses. 

Thank you for your comments. 

Xi Chen

Reviewer 3 Report

The reviewed study extends previous applications of DMSP OLS nighttime lights data to examine 10 the usefulness of newer VIIRS lights in the estimation of economic activity. This is an important contribution to the field of using remote sensning in social science. Focusing on both US 11 states and metropolitan statistical areas, the study found evidence on that the VIIRS lights are more useful in 12 predicting cross-sectional GDP than predicting time-series GDP data. This result is similar to 13 previous findings for DMSP OLS nighttime lights. The findings will help other researchers in the field to use VIIRS for other studies. The paper are scientific sound with visualisations developed to make the results clear. 

The paper has been in review before and now I recommend the journal to publish.

Author Response

Thank you for your comments. 

Xi Chen

Round 2

Reviewer 2 Report

The manuscript had been improved. Suggest publish in present form.